# Prehospital Time Interval for Urban and Rural Emergency Medical Services: A Systematic Literature Review

**DOI:** 10.3390/healthcare10122391

**Published:** 2022-11-29

**Authors:** Abdullah Alruwaili, Ahmed Ramdan M. Alanazy

**Affiliations:** 1Emergency Medical Services Program, College of Applied Medical Sciences, King Saud bin Abdulaziz University for Health Sciences, Al Ahsa 36428, Saudi Arabia; 2King Abdullah International Medical Research Center, Al Ahsa 36428, Saudi Arabia

**Keywords:** emergency medical services, pre-hospital time interval, response time, quality of care, on-scene time, transport time

## Abstract

The aim of this study was to discuss the differences in pre-hospital time intervals between rural and urban communities regarding emergency medical services (EMS). A systematic search was conducted through various relevant databases, together with a manual search to find relevant articles that compared rural and urban communities in terms of response time, on-scene time, and transport time. A total of 37 articles were ultimately included in this review. The sample sizes of the included studies was also remarkably variable, ranging between 137 and 239,464,121. Twenty-nine (78.4%) reported a difference in response time between rural and urban areas. Among these studies, the reported response times for patients were remarkably variable. However, most of them (number (n) = 27, 93.1%) indicate that response times are significantly longer in rural areas than in urban areas. Regarding transport time, 14 studies (37.8%) compared this outcome between rural and urban populations. All of these studies indicate the superiority of EMS in urban over rural communities. In another context, 10 studies (27%) reported on-scene time. Most of these studies (n = 8, 80%) reported that the mean on-scene time for their populations is significantly longer in rural areas than in urban areas. On the other hand, two studies (5.4%) reported that on-scene time is similar in urban and rural communities. Finally, only eight studies (21.6%) reported pre-hospital times for rural and urban populations. All studies reported a significantly shorter pre-hospital time in urban communities compared to rural communities. Conclusions: Even with the recently added data, short pre-hospital time intervals are still superior in urban over rural communities.

## 1. Introduction

Immediate, essential emergency medical services (EMS) are well-known systems for patients suffering from accidents and other individuals with acute or exacerbating emergency conditions on top of other chronic conditions [1]. These services are critical in saving patients’ lives and enhancing the prognoses of their conditions. In this context, relevant previous studies show that with a lack of EMS-related interventions, affected patients will eventually suffer long-term or short-term outcomes after worsening of their medical condition or traumatic injury [2,3]. On the other hand, research shows that enhancing the quality of immediate pre-hospital care can remarkably decrease the incidence of health-related complications and enhance the prognoses of patients [4,5,6].

Pre-hospital time intervals have a huge impact on patients receiving EMS. Evidence shows that these intervals are remarkably different in urban and rural communities [7]. In this context, different metrics have been proposed in the literature, including response time (which is the time taken from receiving the alarm to arriving on-scene), on-scene time (which is the time taken from arriving on-scene to leaving), transport time (which is the time between leaving the scene and arriving at a specialized management centre), and pre-hospital time (which includes all pre-hospital time intervals combined, i.e., the time taken from receiving the alarm to arriving at a specialized management centre). Previous data suggest that pre-hospital time intervals are better in urban areas than in rural areas [3,8]. Moreover, in a previous review, Cabral et al. [8] concluded that studying response time is essential to improving the integrity of EMS in a healthcare system.

There are many factors to consider when evaluating the success of EMS. These include the availability of necessary services, including appropriate vehicles, socioeconomic factors, access to appropriate materials, well-equipped personnel, and coordination of the response process [9,10]. Accordingly, different studies have investigated the impact of these factors, which can affect EMS in rural and urban communities [7,11,12,13,14]. For instance, previous investigations show that patients living in rural countries have longer waiting times for ambulances [12,15,16,17,18]. This is usually associated with severe health-related adverse events and worsens their survival rates [19]. Accordingly, healthcare authorities should enhance the quality of care for this group by considering the delivery of adequate and timely medical services regardless of geographical difficulties [16,20].

Many recent studies have been published to provide more data on whether rural pre-hospital time intervals are comparable to those of urban communities [7,12,14,21,22,23]. Moreover, it is logical that initial management guidelines have remarkably changed since previous data were published. This indicates the need to assess current pre-hospital time intervals, which can help healthcare authorities plan adequate interventions and enhance their EMS and patients’ outcomes. However, no cumulative evidence exists regarding the differences in these metrics and intervals between urban and rural communities. Therefore, this systematic review aims to discuss and update the current knowledge of the differences in pre-hospital time intervals between rural and urban communities regarding EMS.

## 2. Materials and Methods

### 2.1. Study Outcomes and Inclusion Criteria

The main of the present investigation is to compare EMS in rural (all population, housing, and territory not included within an urbanized area or urban cluster) and urban areas (community belonging to, or relating to, a city or town). The terms of comparison will include on-scene time, transport time, response time, and pre-hospital time. Therefore, inclusion criteria include (1) original investigations that (2) compared ground pre-hospital EMS time intervals in rural and urban communities and (3) included patients that required or called EMS secondary to any acute and/or chronic conditions. On the other hand, citations that (1) were not original; (2) included only limited cases (like case reports and case series); (3) did not report either on-scene time, transport time, response time, or pre-hospital time; (4) did not compare between EMS in rural and urban communities; (5) did not include patients in their sample (like studies that obtained their outcomes from surveying healthcare officials and paramedics); (6) were not human studies; (7) were editorials, theses, protocols, commentaries, or reviews; (8) were not published in English; or (9) did not have an accessible full-text were excluded from this study. Finally, definitions of rural and urban settings were established as recognized by the authors of each included study.

### 2.2. Search Strategy

The steps of this systematic review were conducted based on the recommendations of the Preferred Reporting Items for Systematic Reviews and Meta-Analysis (PRISMA). Based on the study outcomes, the relevant keywords were obtained to develop and perform the search strategy, mainly composed of electronic database searching and manual searching, to obtain all relevant investigations. Finally, the following search term was used: (“Emergency Service” OR “Emergency Medical Services” OR “Emergency Medical Technicians” OR “Emergency medicine” OR paramedic* OR ambulance* OR emergency OR trauma OR EMT OR “pre-hospital” OR “out of hospital” OR “EMS”) AND (“Hospitals, Urban” OR “Urban Population” OR “Urban Health Services” OR “Urban Health” OR “Rural Health Personnel” OR “Urban Areas” OR “Hospitals, Rural” OR “Rural Health Services” OR “Rural Areas” OR “Rural Population” OR “Rural Health” OR “Rural Health Centres”) AND (“Response Time” OR “Golden Hour” OR “Duration Time” OR “Transport Time” OR “On-Scene time”). In addition, the following databases were searched: PubMed, Scopus, Web of Science, Cochrane Library and Cochrane Central Register of Controlled Trials (CENTRAL), and Google Scholar in March 2022. It should be noted that only relevant articles published since 1990 were included in this study. This was decided based on a previous meta-analysis by Carr et al. [24], which showed that the quality of EMS care was significantly different in 1990–2005 compared to 1975–1989 in the United States. Accordingly, only articles published since 1990 were included to provide more updated evidence regarding pre-hospital time intervals in rural and urban communities.

### 2.3. Screening Process

After completing the search strategy, articles went through screening and data extraction. The screening was conducted at first by title/abstract, and then by full texts of the included articles. This was performed based on the aforementioned inclusion and exclusion criteria. Next, all the relevant citations were exported from each database into one Endnote library. The program was then used to exclude all the duplicates among these databases to prevent overlapping. The remaining citations were then exported into a standardized Excel sheet designed to fit these articles, based on their titles, authors, DOIs, URLs, and abstracts, to facilitate the screening process. Each step was conducted by at least two reviewers who independently reviewed each article to judge its relevance to the intended outcomes of this study. Finally, a public discussion resolved each disagreement under the supervision of the senior author, who was consulted whenever needed. Before excluding studies published in English or with no available full texts, members searched for relevant data regarding these articles and contacted their authors. Otherwise, these were excluded.

### 2.4. Data Extraction

This step was also conducted by at least two reviewers, similar to the previous step. The extraction sheet was designed by a senior author who conducted a pilot version of extracting relevant data from some included studies to check the suitability of the sheet before going through the extraction process. The sheet was mainly composed of three main parts: a part for baseline characteristics of included studies and their populations, another part for outcomes, and a third part that was particular to quality assessment. Extracted baseline characteristics included the first author’s last name and the year of publication as the reference for the included study, the study design, the data collection process, the sample size, and the age and gender of the included populations. On the other hand, the extracted outcomes include response time, on-scene time, transport time, and pre-hospital time for both rural and urban settings, as well as the significance of each variable and authors’ conclusions. The third part of the sheet included the domains of the quality assessment tools, which will be discussed in the following section.

### 2.5. Quality Assessment

This step was conducted alongside the extraction process, and the assessment process was similarly performed by the study members. For assessing the quality of included observational studies, the National Institutes of Health Quality Assessment Tool was used. In addition, the Cochrane Collaboration’s proposal to assess the risk of bias (RoB 2) for clinical trials was also used [25].

## 3. Results

### 3.1. Search Results

After performing the comprehensive database search, 2445 relevant citations were found since 1990. Endnote was used to remove all potential duplicates and managed to find and exclude 1657 duplicates among the different databases. After title/abstract screening of the remaining citations (n = 788), the full texts of relevant articles (n = 84) were also reviewed. Finally, 26 articles and another 11 relevant articles (obtained via manually searching references of relevant reviews and similar investigations) were included. These steps are summarized in the PRISMA flow chart in Figure 1.

### 3.2. Characteristics of Included Studies

In total, 37 articles were ultimately included in this review. These studies were published between 1991 and 2022. Most studies (n = 17, 45.95%) were conducted in the United States. In contrast, others were conducted in Finland (n = 3, 8.1%), Ireland (n = 3, 8.1%), Sweden (n = 2, 5.4%), the Kingdom of Saudi Arabia (n = 2, 5.4%), Qatar (n = 2, 5.4%), Poland, Denmark, Australia, Scotland, Norway, Taiwan, Iran, and Spain (n = 1, 2.7% each). In addition, the study design for almost all studies (n = 35, 94.6%) was observational, while only two (5.4%) were randomized clinical trials. Twenty-five studies (67.67%) depended on retrospective data collection, while the rest (n = 12, 32.4%) included patients prospectively. The sample size of included studies was also remarkably variable, being highest in the study by Byrne et al. [18] (n = 239,464,121) and lowest in that by Layon et al. [26] (n = 137). However, some studies did not report their total sample size. After assessing the quality of the included studies, nine studies scored 6, nine scored 8, ten scored 7, and six scored 9. Moreover, the assessment of bias in the two trials showed that they had a low risk of bias. These characteristics, together with other variables, are detailed in Table 1.

### 3.3. Study Outcomes

Among the 37 studies in this review, 29 (78.4%) reported a difference in response time between rural and urban areas. Among these studies, the reported response times for their patients were remarkably variable. However, most of them (n = 27, 93.1%) indicate that response times are significantly longer in rural areas than in urban areas. On the other hand, only the study by Grossman et al. [30] showed that mean response time in rural areas is significantly shorter than that in urban areas (7 versus 13.6 min, *p*-value < 0.0001). Moreover, Stripe and Susman [46] reported that response times in both types of communities were similar (Table 2). Regarding transport time, 14 studies (37.8%) compared this outcome between rural and urban populations. All of these studies indicate the superiority of EMS in urban over rural communities. However, the reported mean transport times are also variable among these studies (Table 3). In another context, 10 studies (27%) reported on-scene time. Most of these studies (n = 8, 80%) reported that the mean on-scene time for their populations is significantly longer in rural areas than in urban areas. On the other hand, two studies (20%) reported that on-scene time is similar in urban and rural communities (Table 4). Finally, only eight studies (21.6%) reported pre-hospital time fo r rural and urban populations. All of these studies reported that pre-hospital time is significantly shorter in urban communities than in rural communities. The different pre-hospital time values of each study are detailed in Table 5.

## 4. Discussion

The present systematic review provides cumulative evidence from the relevant studies in the literature regarding the differences between EMS in rural and urban communities in terms of response, transport, and on-scene time. The current findings indicate the superiority of EMS services within urban communities, as most studies indicate that response, transport, and on-scene times are significantly shorter for patients in these areas.

These findings are similar to the results of the previous systematic review by Alanazy et al. [50]. However, the current study provides more updated evidence by including more relevant recent investigations. Moreover, the meta-analysis by Carr et al. also demonstrated that ground urban EMS services are superior to ground rural EMS ones in terms of different pre-hospital time intervals. This indicates the superiority of EMS in urban settings over rural settings, indicating the need to enhance the quality of EMS in the latter settings. However, it should also be noted that the pre-hospital time intervals are remarkably variable among the different studies in the literature, indicating remarkable heterogeneity among these studies and the need for future relevant investigations.

Various factors can contribute to these differences. These include geographical distance (which is usually longer in rural settings); the number and type of available ambulances; the location, number, and preparedness of healthcare facilities; EMS workload, which could determine ambulance queue and the efficiency of the dispatch centre in dispatching ambulances; and transport infrastructure [29,33,51]. These factors can significantly impact response time. Therefore, attempts should be made to enhance the response process. This can be achieved by enhancing the aforementioned factors in rural areas to bring them to a standardized level similar to that of urban areas and improving the quality of EMS by providing well-trained personnel and well-equipped ambulances, according to previous investigations [49,52,53].

However, it should be noted that a position paper in 2003 by the National Association of EMS Physicians recommended that transport and response time intervals have regional variations [54]. Therefore, no specific standards can be applied to adjust these intervals, and they should be determined locally. This encourages conducting further studies per country to help healthcare officials enhance their interventions. In this context, a previous study indicated that using firefighters as medical first responders enhances pre-hospital time and patient outcomes in Sweden [55]. Moreover, in South Korea, the National Ambulance Service suggested that EMS for specific events (like cardiac arrest) might be conducted by moderate care ambulances and non-front-line officers to reduce response time, as such events might not require high levels of training and management resources, which might save time in those situations [56]. Moreover, an RCT by Pappinen et al. [42] compared response times for community first-response models with 1–3 responders and the fire department model. The authors reported that community first-response models might have reduced pre-hospital time compared to the fire department model when emergency vehicles are not used in these events. However, the authors demonstrated that these favourable events are insignificant in rural communities.

In addition, it might be controversial among some studies whether the reduced pre-hospital time is beneficial for patients with traumatic injuries. For instance, the degree of stabilizing traumatized patients in pre-hospital settings is debatable and might impact the outcomes. Accordingly, this might impact on-scene time. Only Grossman et al. [30] reported that rural patients had a shorter mean transport time than urban patients with major traumas. However, establishing a comparison between these patients and other patients with other presenting aetiologies is difficult because of the lack of adequate data in the current literature. It should be noted that the manual by the American College of Surgeons Advanced Trauma Life Support supports reducing on-scene time and rapidly transporting patients to trauma centres [57]. In this context, different relevant studies indicated that these approaches could significantly enhance patients’ outcomes [30,58,59,60,61].

Many studies in the literature indicated socio-spatial disparities in having access to EMS in rural and urban settings [50,62,63,64,65,66,67]. These studies also highlighted the impact of these disparities on patients’ outcomes and mortality rates following various events, like cardiac arrest, stroke, and trauma, and the results favour patients within urban communities [50,68,69]. However, socioeconomic and demographic disparities were also significant variables affecting these outcomes. For instance, previous studies showed that populations with limited resources and foreign individuals usually have limited access to specialized trauma centres [70,71]. In this context, residential discrimination might limit healthcare accessibility to certain ethnic groups and minorities more than others [71,72,73,74]. In addition, reduced levels of accessibility might also be associated with old age, not having medical insurance, and low income levels [64,71,72,75,76].

Although this study included many relevant investigations, there are limitations affecting the interpretation of the current results. The main limitation would be the absence of adjustment of variables that might affect EMS regardless of the geography or distribution of patients. These factors mainly include the probability of patients to seek EMS, such as calling the ambulance, and defining personnel involved in delivering EMS, which might significantly impact the outcomes. For instance, evidence shows that having limited knowledge about whether some symptoms are indicative of a need to call an ambulance or not might hinder EMS. In addition, the probability of calling an ambulance might also be impacted by the patient’s perception of the healthcare system, having social support, self-reported quality of life, and anxiety or loneliness. Another limitation to this study is that we did not investigate the impact of EMS on patients’ outcomes, which is a significant domain that might determine the quality of EMS. Therefore, we encourage future investigations to consider this limitation.

Finally, although studies from different countries were included, evidence from these countries still needs further support by additional investigations to elaborate on the factors that might affect EMS-related patients’ outcomes in rural settings. Furthermore, the absence of adjustment of variables and other co-founders might affect EMS regardless of the geography or distribution of patients. These factors mainly include the probability of patients to seek EMS, such as calling an ambulance, and defining personnel involved in delivering EMS, which might impact outcomes. In addition, evidence from low-to-middle income countries is still lacking; therefore, future investigations from these countries are encouraged to provide more insight regarding their EMS.

## 5. Conclusions

The present systematic review provides cumulative global evidence about the differences in the quality of EMS in rural and urban communities in terms of pre-hospital time intervals. Despite many recently published studies comparing the differences in pre-hospital time intervals between rural and urban communities, the current findings indicate the superiority of urban EMS over rural services, as affected patients in these areas usually have lower response, on-scene, and transport times. This, according to previous studies, is usually associated with better outcomes and favourable prognoses. However, it should be noted that it is difficult in the current study to determine a standardized pre-hospital time interval because of the huge variations among the included global studies. Therefore, future studies are needed to investigate the current limitations and enhance the quality of EMS, particularly for patients living in rural communities.

## Figures and Tables

**Figure 1 healthcare-10-02391-f001:**
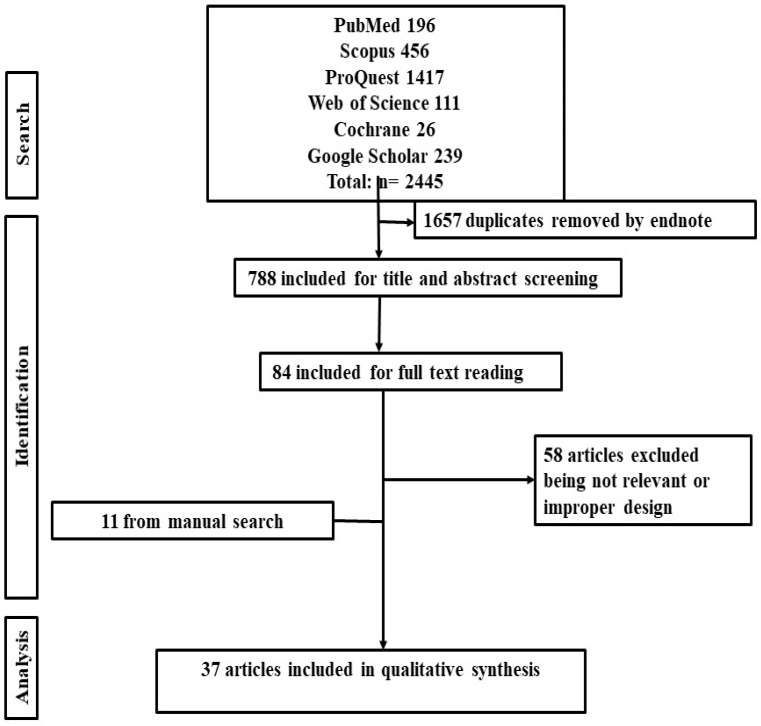
PRISMA flow diagram showing the search strategy of the current systematic review.

**Table 1 healthcare-10-02391-t001:** Baseline characteristics and quality assessment results of the included studies in this review.

Reference	Country	Study Design	Data Collection	Mean Age (SD)–Years	Sample Size	Type of Patient	Conclusion	QA
Adeyemi et al. 2021 [12]	USA	Observational	Prospective	25–49	3108	Crash patients	Favours urban	9
Aftyka et al. 2014 [15]	Poland	Observational	Retrospective	-	1624	ER patients	Favours urban	8
Alanazy et al. 2020 [16]	KSA	Observational	Prospective	42.75/39.72	800	ER patients	Favours urban	8
Al-Thani et al. 2021 [17]	Qatar	Observational	Retrospective	30.9 ± 15.8	1761	Trauma patients	Favours urban	10
Ashburn et al. 2022 [27]	USA	Observational	Retrospective	62 (IQR 50–75)	428,054	Acute cardiac patients	Favours urban	9
Breen et al. 2000 [28]	Ireland	Observational	Prospective	-	3351	ER patients	Favour urban	6
Byrne et al. 2019 [18]	USA	Observational	Prospective	-	239,464,121	ER patients	Favours urban	7
Cui et al. 2021 [22]	USA	Observational	Retrospective		4667	Acute coronary syndrome	Favours urban	6
Cui et al. 2019 [21]	USA	Observational	Retrospective	63.1 (SD, 14.8)	1,672,893	Acute chest pain	Favours urban	8
Fatovich et al. 2011 [11]	Australia	Observational	Retrospective	40.1 ± 22.6	3333	Major trauma patients	Favours urban	7
Gonzalez et al. 2009 [13]	USA	Observational	Retrospective	-	45,763	Crashed patients	Favours urban	7
Gonzalez et al. 2006 [29]	USA	Observational	Retrospective	-	6443	Crashed patients	Favours urban	6
Grossman et al. 1997 [30]	USA	Observational	Prospective	-	459	Major trauma patients	Favours urban	7
Hashtarkhani et al. 2021 [14]	Iran	Observational	Retrospective	43.6 (SD = 22)	224,355	ER patients	Favours urban	9
Hassler et al. 2021 [7]	Sweden	Observational	Retrospective	-	-	ER patients	Favours urban	8
Horeczko et al. 2013 [31]	USA	Observational	Retrospective	<18	283,232,058	ER pediatric patients		6
Hsu et al. 2020 [23]	Taiwan	Observational	Retrospective	-	4225	ER patients	Favours urban	9
Layon et al. 2003 [26]	USA	Observational	Retrospective	65.9 ± 17.4	137	Cardiac arrest	Favours urban	6
Lee et al. 2018 [32]	USA	Observational	Retrospective	-	20,100	Crash patients	Favours urban	7
Masterson et al. 2015 [33]	Ireland	Observational	Retrospective	67 (52–78)	1798	Cardiac arrest	Favours urban	6
Mathiesen et al. 2018 [20]	Norway	Observational	Retrospective	-	1138	Cardiac arrest patients	Favours urban	9
McGuffie et al. 2005 [34]	Scotland	Observational	Prospective	50/64	4636	Traumatic patients	Favours urban	7
Mell et al. 2017 [35]	USA	Observational	Retrospective	-	-	ER patients	Favours urban	7
Michael et al. 2021 [36]	USA	Observational	Retrospective	-	266,605	Trauma patients	Favours urban	8
Moafa et al. 2020 [37]	KSA	Observational	Retrospective	-	146,639	ER patients	Favours urban	6
Moore et al. 2008 [38]	Ireland	Observational	Prospective	67.9 (15.1)	-	Cardiac arrest	Favours urban	7
Morales-Gabardino et al. 2021 [39]	Spain	Observational	Retrospective	-	5572	Trauma patients	Favours urban	8
Newgard et al. 2016 [40]	USA	Observational	Prospective	51.6 ± 26.1	53487	Traumatic patients		6
Nordberg et al. 2015 [41]	Sweden	Observational	Prospective	77/72	2513	Cardiac arrest	Favours urban	7
Pappinen et al. 2021 [42]	Finland	RCT	Retrospective	-	50,000	Trauma patients	Favours urban	Low
Peters et al. 2021 [43]	USA	Observational	Retrospective	-	64,489	ER patients	Favours urban	8
Raatiniemi et al. 2015 [44]	Finland	Observational	Retrospective	33 (20–55)	472	Traumatic patients	Favours urban	8
Sorensen et al. 2010 [45]	Denmark	Observational	Prospective	56 to 79	759	Myocardial infarction	Favours urban	6
Stripe and Susman 1991 [46]	USA	Observational	Prospective	-	-	ER patients	Favours urban	6
Varjoranta et al. 2019 [47]	Finland	Observational	Retrospective	68 (IQR 59, 77)	232	Stroke patients	Favours urban	8
Vukmir et al. 2004 [48]	USA	RCT	Prospective	>18	874	Cardiac arrest	Favours urban	Low
Wilson et al. 2018 [49]	Qatar	Observational	Retrospective	-	394	Traumatic patients	Favours urban	7

RCT; randomized clinical trial; ER; emergency room; USA; United States of America; KSA; Kingdom of Saudi Arabia; QA: Quality assessment by the National Institutes of Health Quality Assessment Tool for observational studies and the Cochrane Collaboration’s proposal to assess the risk of bias (RoB 2) for clinical trials. SD; standard deviation. IQR; interquartile range.

**Table 2 healthcare-10-02391-t002:** Differences between rural and urban areas regarding response time.

Reference	Response Time (Mean)			Conclusion
	Rural	Urban	*p*-Value	
Adeyemi et al. 2021 [12]	19.7	11.1	-	Favours urban
Aftyka et al. 2014 [15]	13.3	7.7	<0.00001	Favours urban
Alanazy et al. 2020 [16]	22	15	<0.001	Favours urban
Al-Thani et al. 2021 [17]	6 * (IQR 4–10)	7 * (IQR 4–10)	0.25	Favours urban
Ashburn et al. 2022 [27]	4.36 longer	shorter	-	Favours urban
Breen et al. 2000 [28]	Longer	Shorter	-	Favours urban
Byrne et al. 2019 [18]	Longer	Shorter	-	Favours urban
Cui et al. 2021 [22]	10–11 longer	shorter	-	Favours urban
Cui et al. 2019 [21]	8 * (IQR 5–13)	7 * (IQR 5–10)	-	Favours urban
Gonzalez et al. 2009 [13]	10.67	6.5	<0.0001	Favours urban
Gonzalez et al. 2006 [29]	13.9	11.2	<0.0002	Favours urban
Grossman et al. 1997 [30]	7	13.6	<0.0001	Favours rural
Hashtarkhani et al. 2021 [14]	12.2	2.1		Favours urban
Hassler et al. 2021 [7]	12.2	7.1	<0.01	Favours urban
Hsu et al. 2020 [23]	Longer	Shorter	<0.001	Favours urban
Lee et al. 2018 [32]	7.1 (11.3)	3.7 (8.5)	-	Favours urban
Masterson et al. 2015 [33]	8 min longer (9%)	8 min longer (33%)	<0.001	Favours urban
Mathiesen et al. 2018 [20]	11 * (IQR 7–16)	9 * (IQR 7–12)	<0.001	Favours urban
Mell et al. 2017 [35]	14.5 (9.5)	7.0 (4.4)	-	Favours urban
Moafa et al. 2020 [37]	20.2 *	15.2 *	<0.001	Favours urban
Moore et al. 2008 [38]	9 11	5–7	-	Favours urban
Morales-Gabardino et al. 2021 [39]	18 (12.6)	10.7 (7.3)	<0.001	Favours urban
Pappinen et al. 2021 [42]	15	1.6	-	Favours urban
Peters et al. 2021 [43]	7.5	5.9	<0.001	Favours urban
Sorensen et al. 2010 [45]	9 min longer	9 min less	<0.001	Favours urban
Stripe and Suaman 1991 [46]	Similar		-	No significance
Varjoranta et al. 2019 [47]	12 * (IQR8, 22)	8 * (IQR 6, 10)	<0.001	Favours urban
Vukmir et al. 2004 [48]	10.6	8.7	0.0002	Favours urban
Wilson et al. 2018 [49]	6.22 *	5.15 *	-	Favours urban

* Represented by median.

**Table 3 healthcare-10-02391-t003:** Differences between rural and urban areas regarding transport time.

Reference	Transport Time (Mean)			Conclusion
	Rural	Urban	*p*-Value	
Ashburn et al. 2022 [27]	0.62 longer	Shorter	-	Favours urban
Byrne et al. 2009 [18]	Longer	Shorter	-	Favours urban
Cui et al. 2021 [22]	Longer	Shorter	-	Favours urban
Cui et al. 2019 [21]	15 * (IQR 7–28)	12 * (IQR 8–19)	-	Favours urban
Fatovich et al. 2011 [11]	11.6 h	59	<0.0001	Favours urban
Gonzalez et al. 2009 [13]	12.45	7.43	<0.0001	Favours urban
Grossman et al. 1997 [30]	17.2	8.2	<0.0001	Favours urban
Hashtarkhani et al. 2021 [14]	20.3	11.2		Favours urban
Hassler et al. 2021 [7]	28.5	7.98	<0.01	Favours urban
Lee et al. 2018 [32]	41.8 (20.5)	28.7 (16.3)	-	Favours urban
Michael et al. 2021 [36]	1.8 *	0.9 *	-	Favours urban
Morales-Gabardino et al. 2021 [39]	45.2 (25)	13.2 (11.7)	0.009	Favours urban
Varjoranta et al. 2019 [47]	65 * (IQR 46, 94)	15 * (IQR 12, 20)	<0.001	Favours urban
Vukmir et al. 2004 [48]	45.8	39.1	0.00005	Favours urban

* Represented by median.

**Table 4 healthcare-10-02391-t004:** Differences between rural and urban areas regarding on-scene time.

Reference	On-Scene Time (Mean)			Conclusion
	Rural	Urban	*p*-Value	
Al-Thani et al. 2021 [17]	23 * (IQR 15–37)	19.5 * (IQR 13–28)	0.001	Favours urban
Byrne et al. 2019 [18]	Longer	Shorter	-	Favours urban
Cui et al. 2021 [22]	10–11 longer	Shorter	-	Favours urban
Cui et al. 2019 [21]	16 * (IQR 12–20)	15 * (IQR 11–20)	-	Favours urban
Gonzalez et al. 2009 [13]	18.87	10.83	<0.0001	Favours urban
Gonzalez et al. 2006 [29]	16.1	11.6		Favours urban
Grossman et al. 1997 [30]	21.7	18.7		Favours urban
Hashtarkhani et al. 2021 [14]	11.2	11.2		Similar
Lee et al. 2018 [32]	14 (10.3)	7.8 (6.1)	-	Favours urban
Varjoranta et al. 2019 [47]	20 (IQR 14, 27)	19 (IQR 14, 23)	0.2	No significance

* Represented by median.

**Table 5 healthcare-10-02391-t005:** Differences between rural and urban areas regarding pre-hospital time.

Reference	Pre-Hospital Time (Mean)			Conclusion
	Rural	Urban	*p*-Value	
Alanazy et al. 2020 [16]	62	43	<0.001	Favours urban
Al-Thani et al. 2021 [17]	87 * (IQR 67–112)	64 * (IQR 49–80)	0.001	Favours urban
Ashburn et al. 2022 [27]	16.67 longer	Shorter	-	Favours urban
Gonzalez et al. 2009 [13]	42	24.8	<0.0001	Favours urban
McGuffie et al. 2005 [34]	Longer	Shorter	<0.001	Favours urban
Moafa et al. 2020 [37]	79.1 *	77.5 *	<0.001	Favours urban
Nordberg et al. 2015 [41]	longer	Shorter	-	Favours urban
Raatiniemi et al. 2015 [44].	longer	Shorter	-	Favours urban

* Represented by median.

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
