# Peer review of "Prehospital Time Interval for Urban and Rural Emergency Medical Services: A Systematic Literature Review"

_healthcare, 2022, doi:10.3390/healthcare10122391_

Round 1
Reviewer 1 Report (New Reviewer)
Abstract
These sentences are not clear
“All of these studies reported that pre-hospital time is significantly shorter in urban than rural 24 communities. Conclusion: Even with the recently-added data, pre-hospital time intervals are still 25 superior in urban over rural communities.”
General comment:
The research is well done as a methodology, but I think I have some concerns.
I think that try to do a systematic review on this topic: I think the geography of each study is critical and for the healthcare system.
Furthermore, time is important, but it is linked to the hub/spoke network and expertise of each center. The hub/spoke network could play an exciting role, especially in case of mass gatherings or disaster.
The results favor urban are predictable: so it will be interesting to measure the outcome for the same disease with the same triage.
Really: the research is well done as a methodology, but I think it does not add anything more to medical literature.
It should discuss better some problems and possible solutions
Author Response
Referee 1
Comment: Abstract, these sentences are not clear “All of these studies reported that pre-hospital time is significantly shorter in urban than rural 24 communities. Conclusion: Even with the recently-added data, pre-hospital time intervals are still 25 superior in urban over rural communities.”
Response: Thank you for your suggestion, we edited this sentence to be clearer.
General comment: The research is well done as a methodology, but I think I have some concerns. I think that try to do a systematic review on this topic: I think the geography of each study is critical and for the healthcare system. Furthermore, time is important, but it is linked to the hub/spoke network and expertise of each center. The hub/spoke network could play an exciting role, especially in case of mass gatherings or disaster. The results favor urban are predictable: so it will be interesting to measure the outcome for the same disease with the same triage. Really: the research is well done as a methodology, but I think it does not add anything more to medical literature. It should discuss better some problems and possible solutions
Response: Thank you so much for your comment, our primary aim was to outline the differences in pre-hospital time intervals between rural and urban communities regarding EMS. We focused on response time, on-scene time, transport time, and pre-hospital time, however, we could not sub-analyze other problems due to the limitation of our included studies discussing other problems and solutions.
Reviewer 2 Report (New Reviewer)
This study analyzes systematic review on the differences in pre-hospital time intervals between rural and urban communities regarding emergency medical services (EMS).
The papers are properly analysed after performing the comprehensive database search.
Please fill in the bibliological information appropriately in the References.
Author Response
Referee 2
Comment: This study analyzes a systematic review of the differences in pre-hospital time intervals between rural and urban communities regarding emergency medical services (EMS). The papers are properly analysed after performing the comprehensive database search. Please fill in the bibliological information appropriately in the References.
Response: Thank you for your valuable input. We updated our citation style and the bibliopolical information in the references according to Chicago-MDPI style.
Reviewer 3 Report (New Reviewer)
This article about EMS system is quite interesting.
But the meta analysis would not be adjustable for the aim of the study.
You mentioned that it is difficult in the current study to determine a standardized prehospital time interval because of the huge variations among the included global studies.
The main limitation would be the absence of adjustment of variables that might affect EMS regardless of the geography or distribution of patients. These factors mainly include the probability of patients to seek EMS, such as calling the ambulance, and defining personnel involved in delivering EMS, which might significantly impact the outcomes.
Therefore, it might be useless to determine difference of Prehospital Time Interval for Urban and Rural Emergency Medical Services globally.
Author Response
Referee 3
Comments and Suggestions for Authors: This article about EMS system is quite interesting. But the meta analysis would not be adjustable for the aim of the study. You mentioned that it is difficult in the current study to determine a standardized prehospital time interval because of the huge variations among the included global studies. The main limitation would be the absence of adjustment of variables that might affect EMS regardless of the geography or distribution of patients. These factors mainly include the probability of patients to seek EMS, such as calling the ambulance, and defining personnel involved in delivering EMS, which might significantly impact the outcomes. Therefore, it might be useless to determine difference of Prehospital Time Interval for Urban and Rural Emergency Medical Services globally.
Response: Thank you so much for your valuable input. We added this point in our limitation.
Reviewer 4 Report (New Reviewer)
Review of the paper: healthcare-1914100:
Pre-hospital Time Interval for Urban and Rural Emergency Medical Services: a systematic review
In this paper, the author (s) are reviewing the existing literature covering the pre-hospital Time Interval for Urban and Rural Emergency Medical Services. A clear methodology is presented for the selection of relevant related publications. The classification and the presentation of the content of these papers are made following some reliable standards. The presented content of this paper is helpful for the interested healthcare community and scientific researchers.
Based on the above comments, I recommend the acceptance of the paper provided that the following minor revisions are addressed.
· Abstract: Define “n” in “them (n = 27, 93.1%)”.
· Abstract: The claim “ two studies (20%)” is incorrect.
· Keywords: Remove “ EMS” keyword since it exists already in the previous keyword “ Emergency medical services”.
· Line 70: At the end of the introduction include the organization of the paper.
· Line 152: replace “Qaultiy” with “Quality”.
· Line 152: More explanations are required for the meaning and how to calculate “Quality assessment”.
· Table 1: Define “IQR” before using it in tables.
Table 1: Define “SD” before using it in tables.
Author Response
Referee 4
Comment: Abstract: Define “n” in “them (n = 27, 93.1%)”.
Response: Thank you for your notice, we defined that abbreviation.
Comment: Abstract: The claim “ two studies (20%)” is incorrect.
Response: Thank you so much for your notice, we corrected that mistake.
Comment: Keywords: Remove “ EMS” keyword since it exists already in the previous keyword “ Emergency medical services”.
Response: Thank you so much for your notice, we removed that keyword.
Comment: Line 152: replace “Qaultiy” with “Quality”.
Response: Thank your for your comment, we corrected this typo.
Comment: Line 152: More explanations are required for the meaning and how to calculate “Quality assessment”.
Response: Thank you for your notice, we explained our methods to assess quality assessment through using two tools scoring system; the National Institutes of Health Quality Assessment Tool and the Cochrane Collaboration tool
Comment: Table 1: Define “IQR” before using it in tables.
Response: Thank you for your comment, we defined this abbreviation.
Comment: Table 1: Define “SD” before using it in tables.
Response: Thank you for your comment, we defined this abbreviation.
Round 2
Reviewer 3 Report (New Reviewer)
I think that it is not enough just to add limitations to method problem.
This article about EMS system is quite interesting. But the meta analysis would not be adjustable for the aim of the study.You mentioned that it is difficult in the current study to determine a standardized prehospital time interval because of the huge variations among the included global studies. The main limitation would be the absence of adjustment of variables that might affect EMS regardless of the geography or distribution of patients. These factors mainly include the probability of patients to seek EMS, such as calling the ambulance, and defining personnel involved in delivering EMS, which might significantly impact the outcomes. Therefore, it might be useless to determine difference of Prehospital Time Interval for Urban and Rural Emergency Medical Services globally.
Author Response
Dear
We really appreciate your efforts in improving our paper.
We understand that adding this part to the limitation is not making much difference. Thus, we would kindly ask for your advice for suggestions on how to improve it further.
Thank you so much for your time given.
This manuscript is a resubmission of an earlier submission. The following is a list of the peer review reports and author responses from that submission.
Round 1
Reviewer 1 Report
The authors provide a compelling and well-written piece of review demonstrating the difference in pre-hospital intervals between rural and urban communities regarding emergency medical services (EMS). This is certainly key in the public health system for EMS. However, there are recommendations made for authors to consider in improving their manuscript, which are listed following:
Introduction
The introduction section should be reconsidered to clearly point out the research problems and the significance. Why is it important to investigate the difference in pre-hospital intervals between rural and urban communities regarding emergency medical services (EMS)? A clear research significance and in-depth analysis should be provided.
Data and methods
In the methodology section, the authors should provide definitions of the on-scene time, transport time, and response time. Why they can be used to demonstrate the difference in pre-hospital intervals between rural and urban communities regarding emergency medical services (EMS)?
Results and discussion
The authors do a good job of explaining in detail their results.
Minor comments in conclusion:
I can not understand the meaning of the “consent information” statement in the conclusion section (lines 284 – 286).
Author Response
Reviewer #1
The authors provide a compelling and well-written piece of review demonstrating the difference in pre-hospital intervals between rural and urban communities regarding emergency medical services (EMS). This is certainly key in the public health system for EMS. However, there are recommendations made for authors to consider in improving their manuscript, which are listed following:
Response
We would like to thank the reviewer for his/her thoughtful comments that have helped to enhance the quality of this research. All the changes are highlighted using track changes.
Introduction
The introduction section should be reconsidered to clearly point out the research problems and significance. Why is it important to investigate the difference in pre-hospital intervals between rural and urban communities regarding emergency medical services (EMS)? A clear research significance and in-depth analysis should be provided.
Response
We agree with you and edited the introduction section accordingly.
Data and methods
In the methodology section, the authors should provide definitions of the on-scene time, transport time, and response time. Why they can be used to demonstrate the difference in pre-hospital intervals between rural and urban communities regarding emergency medical services (EMS)?
Response
Thanks for your sharp notice. We agree with you and added the definitions of these terms where they were first mentioned (in the introduction section).
Results and discussion
The authors do a good job of explaining in detail their results.
Response
Thank you.
Minor comments in conclusion:
I can not understand the meaning of the “consent information” statement in the conclusion section (lines 284 – 286).
Response
Thank you. We agree with you and removed this part.
Reviewer 2 Report
The paper is a systematic review of literature following the PRISMA recommendations. While I believe that this topic definitely deserves to have space in peer-review journals and I therefore congratulate the authors, I have some concerns that in my vision preclude the publication of this review in the absence of significant major reviews.
Introduction
The aim of this paper is to analyze differences in prehospital time intervals between rural and urban communities. Therefore, the authors are focusing on prehospital transportation services worldwide in the attempt to provide an overview of the one of the main barriers in accessing emergency healthcare services, that is geographic accessibility (the others being affordability, availability and acceptability). I believe that the introduction would benefit a complete revision in order to better clarify the focus of the systematic review and the existing gap in knowledge. I struggled to identify the rationale behind this work, and I would suggest to better explain the existing evidence correlating patients’ outcome to prehospital time (from my experience, majority of literature focuses on out of hospital cardiac arrest, very limited evidence exists for trauma patients).
In 2018 Cabral et al published a systematic review on EMS response time worldwide, I believe that this paper should be cited and used as reference both in the introduction and in the discussion section. (Cabral, E., Castro, W., Florentino, D., Viana, D. A., Costa Junior, J., Souza, R. P., Rêgo, A., Araújo-Filho, I., & Medeiros, A. C. (2018). Response time in the emergency services. Systematic review. Acta cirurgica brasileira, 33(12), 1110–1121. https://doi.org/10.1590/s0102-865020180120000009)
Detailed comments:
· Page 1 line 33 “Immediate essential emergency medical services (EMS) are well-known techniques…” I would rephrase this sentence as EMS are not “techniques”, they rather are precisely services (or systems) which on a broader perspective encompass three different components: primary emergency care at the community level, emergency care during transportation (that relates also to access) and emergency care at the receiving facility.
· Page 2 line 43 “These include the availability of necessary techniques” also in this case the word “techniques” is a bit odd. I would suggest rephrasing the sentence.
· Page 2 line 44 “socioeconomic factors of the healthcare facilities and patients” is unclear, I would suggest the authors to rewrite the whole sentence.
· Page 2 line 49 “This is usually associated with severe health-49 related adverse events and worsens their survival rates.” Reference missing.
· Page 2 line 61 “Many recent studies have been published in the literature to provide more data on whether pre-hospital time intervals are becoming comparable to those of urban communities.” References to such studies are missing.
· Page 2 line 63 “However, no cumulative evidence exists regarding the difference between these metrics and intervals between urban and rural communities”
Methods
Methods will benefit a more thorough description of the setting. First, is not immediately clear that you are studying EMS systems worldwide. Second, especially in middle-low and low income countries, there is no formal EMS prehospital ambulance system rather many private local or regional systems exist (and it is not clear if you are also including them or only referring to formal prehospital EMS). Third, prehospital operational time should be correctly defined (eg. is it response time the time from the call receiving to the arrival at the scene or it is considered the time from ambulance dispatch to the arrival at the scene?). Fourth, “Finally, defining rural and urban settings is similar to what has been previously reported in the literature within similar studies” this sentence should either be further elaborated or should cite reference to such similar studies.
Detailed comments:
· Page 3 line 100 “This was done based on the previous meta-analysis by Carr et al. (21), which showed that the quality of EMS care was significantly different in 1990-2005 from 1975-1989 in the United States” are we therefore assuming that this is the case also for remaining countries?
Results
For some reasons, the systematic review missed some recent and important articles that would fit inclusion criteria:
Mell HK, Mumma SN, Hiestand B, Carr BG, Holland T, Stopyra J. Emergency Medical Services Response Times in Rural, Suburban, and Urban Areas. JAMA Surg. 2017;152(10):983-984. doi:10.1001/jamasurg.2017.2230
José Antonio Morales-Gabardino, Laura Redondo-Lobato and João Meireles Ribeiro et al. Geographical distribution of emergency services times in traffic accidents in Extremadura. Port J Public Health. Vol. 39(2):78-87. DOI: 10.1159/000519858
ALINIER, G. ., Wilson , P., Reimann , T., & Morris , B. (2021). INFLUENTIAL FACTORS ON URBAN AND RURAL RESPONSE TIMES FOR EMERGENCY AMBULANCES IN QATAR. Mediterranean Journal of Emergency Medicine, (26), 8-13. Retrieved from https://www.mjemonline.com/index.php/mjem/article/view/82
Breen, N., Woods, J., Bury, G., Murphy, A. W., & Brazier, H. (2000). A national census of ambulance response times to emergency calls in Ireland. Journal of accident & emergency medicine, 17(6), 392–395. https://doi.org/10.1136/emj.17.6.392
I would like to understand why such manuscripts have been excluded in the search process.
On a side note, I believe that it is remarkable that no middle-low/low income countries (in which, notoriously, the difference between rural and urban areas are extreme) has been intercepted by the review.
Discussion
Similarly to the introduction section, I believe that the discussion could benefit a complete revision to better structure the flow and logic of the key findings in relation the available literature. For instance, authors begin the discussion introducing the concept of BLS and ALS (4 and 8 minutes) which has never been mentioned before in the paper.
Detailed comments:
· Page 8 line 212: “Moreover, the meta-analysis by Carr et al. also demonstrated that ground urban EMS services are superior to ground rural EMS ones in terms of different pre-hospital time intervals. This indicates the superiority of EMS in urban over rural settings, indicating the need to enhance the quality of care in these settings” I believe that the authors should be more cautious as their analysis only focuses on prehospital times with no reference on the quality of care provided during transport. This is an important aspect that is currently lacking in the paper.
· Page 8 line 217: “Various factors can attribute to these differences. These include geographical distance (which is usually longer in rural settings), the number and type of available ambulances, the location, number, and preparedness of healthcare facilities, and transport infrastructure (31-33).” This list is missing many other two factors that can influence prehospital time, that are 1) the efficiency of the dispatch center in dispatching the ambulances 2) EMS workload that could determine ambulance queue
· Page 8 line 220: “Therefore, attempts should be made to enhance the response process.” The author should further elaborate (also based on evidence in the literature) on the specific strategies that could be put in place to “enhance” the response process.
Finally, I would suggest that the authors get editing help from someone with full professional proficiency in English to proofread the paper.
Author Response
Reviewer #2
The paper is a systematic review of literature following the PRISMA recommendations. While I believe that this topic definitely deserves to have space in peer-review journals and I therefore congratulate the authors, I have some concerns that in my vision preclude the publication of this review in the absence of significant major reviews.
Response
We would like to thank the reviewer for his/her thoughtful comments that have helped to enhance the quality of this research. All the changes are highlighted using track changes.
Introduction
The aim of this paper is to analyze differences in prehospital time intervals between rural and urban communities. Therefore, the authors are focusing on prehospital transportation services worldwide in the attempt to provide an overview of one of the main barriers in accessing emergency healthcare services, that is geographic accessibility (the others being affordability, availability and acceptability). I believe that the introduction would benefit a complete revision in order to better clarify the focus of the systematic review and the existing gap in knowledge. I struggled to identify the rationale behind this work, and I would suggest to better explain the existing evidence correlating patients’ outcome to prehospital time (from my experience, majority of literature focuses on out of hospital cardiac arrest, very limited evidence exists for trauma patients).
In 2018 Cabral et al published a systematic review on EMS response time worldwide, I believe that this paper should be cited and used as reference both in the introduction and in the discussion section. (Cabral, E., Castro, W., Florentino, D., Viana, D. A., Costa Junior, J., Souza, R. P., Rêgo, A., Araújo-Filho, I., & Medeiros, A. C. (2018). Response time in the emergency services. Systematic review. Acta cirurgica brasileira, 33(12), 1110–1121. https://doi.org/10.1590/s0102-865020180120000009)
Response
Thank you. We cited the mentioned paper. We would like to clarify that the current systematic review was only conducted to compare pre-hospital time intervals between rural and urban communities. Accordingly, assessing patients’ outcomes in both communities was not a part of the outcomes for which this review was conducted. However, we agree that such outcomes are very important to assess and would benefit current literature regarding the differences between rural and urban EMS. We called for future systematic reviews to assess these outcomes inn future settings and mentioned this in the limitations section.
Moreover, more than one-third of the included studies included patients with cardiac arrest/acute chest pain. Therefore, we decided not to focus the current data on patients with trauma only since many studies have investigated these conditions.
Detailed comments:
- Page 1 line 33 “Immediate essential emergency medical services (EMS) are well-known techniques…” I would rephrase this sentence as EMS are not “techniques”, they rather are precisely services (or systems) which on a broader perspective encompass three different components: primary emergency care at the community level, emergency care during transportation (that relates also to access) and emergency care at the receiving facility.
Response
We agree with you and edited it accordingly.
Page 2 line 43 “These include the availability of necessary techniques” also in this case the word “techniques” is a bit odd. I would suggest rephrasing the sentence.
Response
We agree with you and edited it accordingly.
- Page 2 line 44 “socioeconomic factors of the healthcare facilities and patients” is unclear, I would suggest the authors to rewrite the whole sentence.
Response
We agree with you and edited it accordingly.
- Page 2 line 49 “This is usually associated with severe health-49 related adverse events and worsens their survival rates.” Reference missing.
Response
We agree with you and edited it accordingly.
- Page 2 line 61 “Many recent studies have been published in the literature to provide more data on whether pre-hospital time intervals are becoming comparable to those of urban communities.” References to such studies are missing.
Response
We agree with you and edited it accordingly.
Methods
Methods will benefit a more thorough description of the setting. First, is not immediately clear that you are studying EMS systems worldwide. Second, especially in middle-low and low income countries, there is no formal EMS prehospital ambulance system rather many private local or regional systems exist (and it is not clear if you are also including them or only referring to formal prehospital EMS). Third, prehospital operational time should be correctly defined (eg. is it response time the time from the call receiving to the arrival at the scene or it is considered the time from ambulance dispatch to the arrival at the scene?). Fourth, “Finally, defining rural and urban settings is similar to what has been previously reported in the literature within similar studies” this sentence should either be further elaborated or should cite reference to such similar studies.
Response
Thank you for your comments. The present study is a systematic review that basically depends on data collection from the relevant studies in the literature. Accordingly, we decided to include any study that compared between pre-hospital time intervals of rural and urban communities regardless of their geographical distribution. Accordingly, some studies (like the USA) might be represented by many studies, as exhibited in Table 1, while many others are not or poorly represented, especially studies from low-to-middle income countries. The importance of this research is to establish a spark that calls for investigating such outcomes by future studies in these countries as we mentioned in the limitation section.
We defined each time interval where they were first mentioned (in the introduction section).
Defining rural and urban communities was established as recognized by the authors of each included study and we edited this part in the methods section.
Detailed comments:
- Page 3 line 100 “This was done based on the previous meta-analysis by Carr et al. (21), which showed that the quality of EMS care was significantly different in 1990-2005 from 1975-1989 in the United States” are we therefore assuming that this is the case also for remaining countries?
Response
We agree with you. However, since most of the included studies were conducted in the USA, and there is a wide geographical distribution in the rest of included studies, we decided to set this study as an example of the inevitable change in EMS, especially as we are talking about decades not a small set of years.
Results
For some reasons, the systematic review missed some recent and important articles that would fit inclusion criteria:
Mell HK, Mumma SN, Hiestand B, Carr BG, Holland T, Stopyra J. Emergency Medical Services Response Times in Rural, Suburban, and Urban Areas. JAMA Surg. 2017;152(10):983-984. doi:10.1001/jamasurg.2017.2230
José Antonio Morales-Gabardino, Laura Redondo-Lobato and João Meireles Ribeiro et al. Geographical distribution of emergency services times in traffic accidents in Extremadura. Port J Public Health. Vol. 39(2):78-87. DOI: 10.1159/000519858
ALINIER, G. ., Wilson , P., Reimann , T., & Morris , B. (2021). INFLUENTIAL FACTORS ON URBAN AND RURAL RESPONSE TIMES FOR EMERGENCY AMBULANCES IN QATAR. Mediterranean Journal of Emergency Medicine, (26), 8-13. Retrieved from https://www.mjemonline.com/index.php/mjem/article/view/82
Breen, N., Woods, J., Bury, G., Murphy, A. W., & Brazier, H. (2000). A national census of ambulance response times to emergency calls in Ireland. Journal of accident & emergency medicine, 17(6), 392–395. https://doi.org/10.1136/emj.17.6.392
I would like to understand why such manuscripts have been excluded in the search process.
Response
Thanks for your sharp notice. I believe we missed these articles in the search process and accordingly they were added to the current review and we edited the manuscript accordingly.
On a side note, I believe that it is remarkable that no middle-low/low income countries (in which, notoriously, the difference between rural and urban areas are extreme) has been intercepted by the review.
Response
We agree with you. However, our results reflect currently available data in the literature. We also called for future investigations in these countries to provide better data.
Discussion
Similarly to the introduction section, I believe that the discussion could benefit a complete revision to better structure the flow and logic of the key findings in relation the available literature. For instance, authors begin the discussion introducing the concept of BLS and ALS (4 and 8 minutes) which has never been mentioned before in the paper.
Response
We agree with you. We decided to remove this part and enhanced the structure of our discussion by starting with our main results followed by interpretation of these finding and comparison with the current literature.
Detailed comments:
- Page 8 line 212: “Moreover, the meta-analysis by Carr et al. also demonstrated that ground urban EMS services are superior to ground rural EMS ones in terms of different pre-hospital time intervals. This indicates the superiority of EMS in urban over rural settings, indicating the need to enhance the quality of care in these settings” I believe that the authors should be more cautious as their analysis only focuses on prehospital times with no reference on the quality of care provided during transport. This is an important aspect that is currently lacking in the paper.
Response
We agree with you. However, we suggested (not concluded) based on our findings that the quality of EMS should be improved in rural settings to enhance their pre-hospital times.
- Page 8 line 217: “Various factors can attribute to these differences. These include geographical distance (which is usually longer in rural settings), the number and type of available ambulances, the location, number, and preparedness of healthcare facilities, and transport infrastructure (31-33).” This list is missing many other two factors that can influence prehospital time, that are 1) the efficiency of the dispatch center in dispatching the ambulances 2) EMS workload that could determine ambulance queue.
Response
Thanks. We edited the discussion accordingly.
- Page 8 line 220: “Therefore, attempts should be made to enhance the response process.” The author should further elaborate (also based on evidence in the literature) on the specific strategies that could be put in place to “enhance” the response process.
Response
Thanks. We edited the discussion accordingly.
Finally, I would suggest that the authors get editing help from someone with full professional proficiency in English to proofread the paper.
Response
Thanks. A native English speaker checked the linguistic quality and proofread the manuscript as suggested.
Reviewer 3 Report
I'm afraid the quality of english language makes it very hard to assess whether the article has merit.
There are serious inconsistencies in the presentation of the results which contradict each other. For example:
Table one includes the conclusion “favours urban” for all 33 studies. However, line 170 – the authors state that only “ 25 [studies] (75.8%) reported the difference in response 170
time between rural and urban areas” This also contradicts Table 2.
The authors conclusions of the cited South Korea study ref 36, seem strange.
I am happy to review the article again after the english language has been revised, but currently it is hard to assess the quality of the article or offer recomenndations.
Author Response
Reviewer #3
We would like to thank the reviewer for his/her thoughtful comments that have helped to enhance the quality of this research. All the changes are highlighted using track changes.
'm afraid the quality of English language makes it very hard to assess whether the article has merit.
There are serious inconsistencies in the presentation of the results which contradict each other. For example:
Table one includes the conclusion “favours urban” for all 33 studies. However, line 170 – the authors state that only “ 25 [studies] (75.8%) reported the difference in response 170
time between rural and urban areas” This also contradicts Table 2.
The authors conclusions of the cited South Korea study ref 36, seem strange.
I am happy to review the article again after the english language has been revised, but currently it is hard to assess the quality of the article or offer recomenndations.
Response
Thanks. A native English speaker checked the linguistic quality and proofread the manuscript as suggested.
Regarding Table 1, “favours urban” cumulatively refers to the authors’ main conclusions that EMS is better in urban than rural communities. So no contraindication between what has been mentioned in this table and others since each other table investigates a single outcome and all studies reported all the mentioned outcomes, so it is logical that the number of studies would be smaller than what has been mentioned in Table 1, which summarizes the baseline characteristics of all included studies.
Regarding Ref. 36, we cited it together with other citations from different geographical regions because our study is not limited to a single country, so we believed that it would benefit our discussion to establish solid and comprehensive evidence.
Reviewer 4 Report
Dear authors,
I congratulate you on your systematic review, which I think is very relevant and highlights the logistical differences in emergency care in urban and rural areas.
Despite this, I believe that there are aspects that need to be improved:
At the beginning of the introduction, EMS care is mentioned in the case of trauma or worsening of a chronic disease, when it is usual to deal with acute pathologies such as heart attacks or strokes. These aspects are covered later and I believe that this can lead to confusion. Moreover, it is not only these situations that are dependent on response times, care and transfer times, as there are others within this group, such as sepsis, for example.
In the introduction, several times are mentioned: response time, on-scene time, transport time and pre-hospital time. It would be appropriate to make it clear what each time corresponds to. Personally, I do not understand what is meant by pre-hospital time, as all the others are pre-hospital time as well.
It is not made clear in the introduction whether it is exclusively ground response or also air response, which would significantly reduce times in rural areas.
Nor is it specified whether the EMS model to be studied is one based on doctors, nurses and technicians or another, since response time is important but so is the quality of care.
If they state that there are hardly any studies comparing response data between urban and rural settings, then a systematic review would not be the most appropriate study, but one that measures these differences.
The methodology, as well as the inclusion and exclusion criteria should be improved, as they are difficult to read.
The search strategy they propose is adequate, but the strategy should be specified for each database.
They could follow the example of this systematic review:https://doi.org/10.3390/jcm10235578
It is necessary to specify in the tables which tool is used to specify the quality of the study, although this data appears in the methodology. I recommend including it.
Looking at the results, there is concern that countries with disparate health systems and different EMSs are being compared, which may represent a significant bias for the study. EMS from countries with similar health systems, geographical conditions and distances to similar ambulances should be compared, and not generalise the times on the scene but individualise them by pathology, as the same time is not consumed with a stroke as with a polytrauma patient.
It is really surprising that there are longer response times in an urban or impossible area. Have you analysed whether this study is well thought out and what is the reason for this time? There are some other surprising results for which no further data are given.
In the conclusions, I do not see the need to ask for informed consent from patients. It is not a conclusion of the study and it is obvious.
I hope my contributions will help you to improve your study.
Author Response
Reviewer #4
I congratulate you on your systematic review, which I think is very relevant and highlights the logistical differences in emergency care in urban and rural areas.
Despite this, I believe that there are aspects that need to be improved:
Response
We would like to thank the reviewer for his/her thoughtful comments that have helped to enhance the quality of this research. All the changes are highlighted using track changes.
At the beginning of the introduction, EMS care is mentioned in the case of trauma or worsening of a chronic disease, when it is usual to deal with acute pathologies such as heart attacks or strokes. These aspects are covered later and I believe that this can lead to confusion. Moreover, it is not only these situations that are dependent on response times, care and transfer times, as there are others within this group, such as sepsis, for example.
Response
Thanks. We edited this part as our study aimed to include all types of patients that required EMS.
In the introduction, several times are mentioned: response time, on-scene time, transport time, and pre-hospital time. It would be appropriate to make it clear what each time corresponds to. Personally, I do not understand what is meant by pre-hospital time, as all the others are pre-hospital time as well.
Response
Thanks. We edited this part according to your suggestions.
It is not made clear in the introduction whether it is exclusively ground response or also air response, which would significantly reduce times in rural areas.
Response
Thanks. We included studies that investigated ground response only.
Nor is it specified whether the EMS model to be studied is one based on doctors, nurses and technicians or another since response time is important but so is the quality of care.
Response
Thanks. We included all studies that reported patients requiring EMS. We believe that EMS is usually conducted by technicians/paramedics since patients are not usually transferred to specialized care centers where they can receive their required care. Accordingly, we believe that included studies did not routinely report this point since it is widely known that technicians/paramedics majorly have this role.
The methodology, as well as the inclusion and exclusion criteria should be improved, as they are difficult to read.
Response
Thanks. We edited this part and hope that it is more comprehensible.
The search strategy they propose is adequate, but the strategy should be specified for each database.
They could follow the example of this systematic review:https://doi.org/10.3390/jcm10235578
Response
Thanks. We agree with you. However, changing the search strategy means that we need to change the current structure of the review. Besides, we believe that the current search strategy is valid as we followed the PRISMA guidelines and it is also similar to many previous studies. However, it is a nice suggestion to be followed in future investigations.
It is necessary to specify in the tables which tool is used to specify the quality of the study, although this data appears in the methodology. I recommend including it.
Response
Thanks. We edited this part as suggested.
Looking at the results, there is concern that countries with disparate health systems and different EMSs are being compared, which may represent a significant bias for the study. EMS from countries with similar health systems, geographical conditions and distances to similar ambulances should be compared, and not generalise the times on the scene but individualise them by pathology, as the same time is not consumed with a stroke as with a polytrauma patient.
Response
Thanks. We agree with you and provided the country and type of patient per each study in the baseline characteristics table so we believe it would provide a better insight to the readers.
It is really surprising that there are longer response times in an urban or impossible area. Have you analysed whether this study is well thought out and what is the reason for this time? There are some other surprising results for which no further data are given.
Response
Thanks. We agree with you. However, no further data or justifications were provided by the authors of this study. We believe, according to some evidence that EMS in rural communities can be enhanced by standardizing the quality of EMS in rural communities similar to that in urban ones.
In the conclusion, I do not see the need to ask for informed consent from patients. It is not a conclusion of the study and it is obvious.
Response
Thanks. We edited this part as suggested.
Round 2
Reviewer 2 Report
Introduction
Although I appreciate the effort of authors in reviewing the introduction section, I still think it lacks clarity and structure. The aim of the manuscript is to provide an overview of overall prehospital times and times intervals (response time, on-scene time and transport time) focusing on the difference of such metric in urban vs rural context worldwide. As such, I would structure the introduction in such a way that the following are covered:
· What is known about the topic: EMS and prehospital times (it should better reflect the context of the subject) in a clear and concise manner.
· What is the gap in knowledge: do the geographical barriers, socioeconomic factors and all the differences existing between urban and rural area impact on prehospital times (step back: is there any difference between urban and rural area?)
· Evidence providing rationale for the study: how does this review improve already existing evidence?
· Study aim/objective/task
Methods & Results
When was the search conducted? This piece of information is extremely important also to understand the reason why some recent manuscripts have not been included in the search and that the review really represents an up to date overview of existing evidence.
Discussion
That there are still some flaws in the discussion structure that affect the readability of the paper. I would suggest to the authors to discuss the results in a more organized manner.
Specific comments:
Page 9 line 224 “The current findings indicate the superiority of EMS services in these outcomes” the word “outcomes” is misleading, I suggest rephrasing.
Page 9 line 234” This can be achieved by enhancing the aforementioned factors in rural areas to come to a standardized level similar to that of urban areas and improving the quality of EMS by providing well-trained personnel and well-equipped ambulances” This paragraph is oddly contradicting the next one, as in its current version it seems a recommendation supported by authors. I suggest rephrasing.
As a general comment, the english language could benefit a second round of proof reading especially in the introduction section.
Author Response
Response to reviewers’ comments
===============================
Reviewer #2
Introduction
Comment: Although I appreciate the effort of authors in reviewing the introduction section, I still think it lacks clarity and structure. The aim of the manuscript is to provide an overview of overall prehospital times and times intervals (response time, on-scene time and transport time) focusing on the difference of such metric in urban vs rural context worldwide. As such, I would structure the introduction in such a way that the following are covered:
- What is known about the topic: EMS and prehospital times (it should better reflect the context of the subject) in a clear and concise manner.
- What is the gap in knowledge: do the geographical barriers, socioeconomic factors and all the differences existing between urban and rural area impact on prehospital times (step back: is there any difference between urban and rural area?)
- Evidence providing rationale for the study: how does this review improve already existing evidence?
Response
We agree with you and edited the introduction section accordingly.
Methods & Results
Comment: When was the search conducted? This piece of information is extremely important also to understand the reason why some recent manuscripts have not been included in the search and that the review really represents an up to date overview of existing evidence.
Response
Thanks for your sharp notice. We agree with you and added it in the methods section.
Discussion
Comment: That there are still some flaws in the discussion structure that affect the readability of the paper. I would suggest to the authors to discuss the results in a more organized manner.
Specific comments:
Page 9 line 224 “The current findings indicate the superiority of EMS services in these outcomes” the word “outcomes” is misleading, I suggest rephrasing.
Page 9 line 234” This can be achieved by enhancing the aforementioned factors in rural areas to come to a standardized level similar to that of urban areas and improving the quality of EMS by providing well-trained personnel and well-equipped ambulances” This paragraph is oddly contradicting the next one, as in its current version it seems a recommendation supported by authors. I suggest rephrasing.
As a general comment, the english language could benefit a second round of proof reading especially in the introduction section.
Response
Thank you. We agree with you and edited the discussion accordingly.
Reviewer 4 Report
Dear authors, thanks for your modifications, but despite this, I think there are aspects that remain unimproved. It remains unclear what prehospital time is. It is not true that the paramedic model is used exclusively in EMS. There are many countries that also employ doctors and nurses. Not specifying it is a bias. Following the PRISMA guide and making a table according to that strategy does not mean that a review is well done. Indeed, PRISMA is widely used, and the search strategy in the databases is also specified on many occasions. This aspect brings enormous quality to reviews. The explanation they offer for less time in rural areas is not very convincing. Aren't there more studies to back it up? I hope you can improve these aspects. All the best. The reviewer.
Author Response
Reviewer #4
Comment: Dear authors, thanks for your modifications, but despite this, I think there are aspects that remain unimproved.
Response
We would like to thank the reviewer for his/her thoughtful comments that have helped to enhance the quality of this research. All the changes are highlighted using track changes.
Comment: It remains unclear what prehospital time is.
Response
Thank you. We edited it.
Comment: It is not true that the paramedic model is used exclusively in EMS. There are many countries that also employ doctors and nurses. Not specifying it is a bias.
Response
We agree with you. However, such data was not provided in included studies. We included this point in the limitations section to be considered by future investigations.
Comment: Indeed, PRISMA is widely used, and the search strategy in the databases is also specified on many occasions. This aspect brings enormous quality to reviews.
Response
Thank you. We agree with you.
Comment: The explanation they offer for less time in rural areas is not very convincing. Aren't there more studies to back it up? I hope you can improve these aspects. All the best. The reviewer.
Response
We agree with you. However, such data remain lacking in the current literature and we tried to include all relevant studies in the literature and called for future studies to assess these concerns at the end of our study.